# An inter-comparison of approaches and frameworks to quantify irrigation from satellite data

Søren Julsgaard Kragh[1], Jacopo Dari[2,3], Sara Modanesi[3], Christian Massari[3], Luca Brocca[3,] Rasmus Fensholt[4], Simon Stisen[1], Julian Koch[1]

[1] Department of Hydrology, Geological Survey of Denmark and Greenland, Copenhagen, 1350, Denmark
[2] Department of Civil and Environmental Engineering, University of Perugia, via G. Duranti 93, 06125 Perugia, Italy
[3] National Research Council, Research Institute for Geo-Hydrological Protection, via Madonna Alta 126, 06128 Perugia, Italy
[4] Department of Geosciences and Natural Resource Management, University of Copenhagen, Copenhagen, 1350, Denmark

*Correspondence to*: Søren Julsgaard Kragh (sjk@geus.dk)

**Abstract**

This study provides the first inter-comparison of different state-of-the-art approaches and frameworks that share a commonality in their utilization of satellite remote sensing data to quantify irrigation at a regional scale. The compared approaches vary in their reliance on either soil moisture or evapotranspiration data, or their joint utilization of both. The two compared frameworks either extract irrigation information from residuals between satellite observations and rainfed hydrological models in a baseline framework or use soil water balance modeling in a soil moisture-based inversion framework. The inter-comparison is conducted over the lower Ebro catchment in Spain where observed irrigation amounts are available for benchmarking. Our results showed that within the baseline framework, the joint approach using both soil moisture and ET remote sensing data, only differed by +37 mm from the irrigation benchmark (922 mm) during the main irrigation season over two years, and by +47 mm and -208 mm for approaches relying solely on soil moisture and ET, respectively. A comparison of the different frameworks showed that the main advantage of the more complex baseline framework was the consistency between soil moisture and ET components within the hydrological model, which made it unlikely that either one ended up representing all irrigation water use. However, the simplicity of the soil moisture-based inversion framework, coupled with its direct conversion of soil moisture changes into actual water volumes, effectively addresses the key challenges inherent in the baseline framework, which are associated with uncertainties related to an unknown remote sensing observation depth and the static depth of the soil layers in a conceptual model. The performance of the baseline framework came closest to the irrigation benchmark and was able to account for the precipitation input, which resulted in more plausible temporal distributions of irrigation than what was expected from the benchmark observations.

## 1 Introduction

As illustrated in the newly updated version of the water cycle by the USGS (Corson-Dosch et al., 2023) the role of irrigation is now manifested as one of the major hydrological fluxes, which in many regions cannot be ignored when resolving the water

balance (Döll and Siebert, 2002). Further, with future projections of climate change, population growth, and food demand (Hunter et al., 2017), irrigation-based agriculture is expected to become an even more vital part of the water cycle that needs

to be managed sustainably to secure global food security and livelihoods for the billions of people living in arid regions (Ferguson et al., 2018; Jain et al., 2021; Mujumdar, 2013). The main obstacle in managing this major flux component is the lack of knowledge about where, when, and how much irrigation is applied.

Large-scale irrigation mapping and quantification from satellite data have commonly been studied by the hydrological community and the number of studies addressing these questions has rapidly increased over the last decade (Massari et al.,

2021). Two of the major branches of proposed frameworks are the soil moisture inversion frameworks (SM-based inversion framework) (Brocca et al., 2018) hypothesizing that irrigation can be estimated by backward calculation of total water input from an inversion of the soil water balance equation by using remote sensing soil moisture observations. The other branch is the baseline frameworks, hypothesizing that differences between purely rainfed hydrological models and satellite remote sensing products of soil moisture and evapotranspiration (ET) are contributed to by a non-precipitation source of water.

The SM-based inversion framework is rooted in the soil moisture to rain algorithm (Brocca et al., 2014) intended to estimate global precipitation from soil moisture satellite observations. The documented alteration of the soil moisture signal by irrigation practices (Filippucci et al., 2020) is leveraged to estimate irrigation as a distinct precipitation component using soil moisture inversion techniques, which is subsequently isolated by subtracting the measured precipitation. The framework has successfully been applied in studies focusing on mapping and quantifying irrigation in various regions as well as on a

global scale, and several studies have been aiming at validating the framework by testing different soil moisture remote sensing products, refining the ET contribution and calibration strategies (Dari et al., 2020, 2021, 2022b, 2023; Filippucci et al., 2020; Jalilvand et al., 2019; Zhang et al., 2022). One of the main challenges within the SM-based inversion framework is to ensure the right magnitude of contributions between soil moisture, drainage, and ET by targeting the precipitation input through calibration. As pointed out by (Jalilvand et al., 2019) and Dari et al. (2022), a simple soil water stress approach to quantify the

actual ET is a crucial part of the framework that potentially can restrain the calibration parameters.

The baseline frameworks can be found in many varieties but overall, they aim to quantify irrigation from soil moisture or ET residuals between retrieved changes from satellite remote sensing models and a rainfed baseline, typically a hydrological model without an explicit representation of irrigation. Lawston et al. (2017) used satellite soil moisture observations to estimate irrigation patterns by comparing soil moisture dynamics of irrigated and rainfed satellite pixels in the western United States.

Similarly, Brombacher et al. (2022) compared irrigated actual ET with a rainfed reference ET through a hydrological similar pixels algorithm in three irrigated regions in Africa, Spain, and Australia. In both studies, the latter acted as a baseline. Zappa et al. (2021) combined a comparison of local and regional surface soil moisture changes with independent ET and drainage loss estimates to quantify irrigation in Germany. Other studies have compared satellite remote sensing models and hydrological models to extract soil moisture changes associated with a non-precipitation source of water to estimate irrigation (Zaussinger

et al., 2019; Zohaib and Choi, 2020). Koch et al. (2020) and (Kragh et al., 2023) further compared remote sensing model retrievals of actual ET from irrigated cropland areas with a rainfed hydrological model calibrated against the remote sensing

model retrieval of actual ET over rainfed cropland (hence with natural vegetation) to counteract the influence of a possible rainfed bias between the remote sensing and hydrological models. The main challenge within this framework relates to the uncertainty of the remote sensing products used and the estimation of a rainfed baseline when combining satellite remote sensing models and hydrological models. Especially the soil moisture-based approaches are faced with the need for bias corrections or rescaling between independent satellite remote sensing and hydrological model estimates to assure coherence between the magnitudes of the rainfed estimates (Reichle and Koster, 2004).

Another branch of irrigation quantification frameworks involves the assimilation of satellite data into hydrological/land surface models. Modanesi et al. (2022) used the Noah-MP model with an irrigation scheme and assimilated vegetation-sensitive Sentinel 1 VH polarization backscatter to improve irrigation simulations. Abolafia-Rosenzweig et al. (2019) proposed an ensemble methodology where satellite soil moisture data was assimilated with a model ensemble forced with precipitation and superimposed irrigation signals to extract the total water input and hence irrigation by subtracting precipitation. Further work has lately been conducted on evaluating the framework sensitivity and incorporating a rainfed bias correction to address the challenge of simulating a correct soil moisture state (Jalilvand et al., 2023), which is a challenge for all frameworks that try to combine satellite data and hydrological models.

An inter-comparison is needed to further advance the fast-evolving development of irrigation quantification methodologies toward more robust estimates by elucidating the assets and shortcomings within the framework structures. However, an in-depth inter-comparison is challenging due to the general absence of in-situ irrigation observations (Massari et al., 2021) needed to go a step beyond the validation of area-averaged (e.g., at the district scale) irrigation estimates. Furthermore, many irrigation studies exhibit a unidirectional emphasis, either relying solely on soil moisture or ET remote sensing data. Consequently, they overlook the potential synergy that can be derived from an integrated approach capable of capturing the characteristics of both equally important components.

This study applies a novel baseline methodology to quantify irrigation from a joint approach using both soil moisture and ET in a dual component calibration for a rainfed model baseline and compares the results with approaches focusing on either soil moisture or ET in isolation within the same framework. This extends to previous work (Koch et al., 2020; Romaguera et al., 2014) and aims to showcase the potential advantages of moving toward joint soil moisture and ET-based approaches. Also, a stand-alone near-surface soil moisture approach within a baseline framework is included aiming at quantifying irrigation solely from soil moisture residuals. The analysis is carried out in a well-studied irrigation hot spot area situated in the lower Ebro catchment in Spain and covers the period 2016 - 2017. The three main objectives of this paper are: 1) to extend an earlier ET-based baseline framework to also include soil moisture, 2) to compare irrigation estimates from a soil moisture-based, ET-based, and joint use of soil moisture and ET satellite datasets to explore any gains, and 3) attempt to compare and validate irrigation estimates from baseline frameworks and SM-based inversion frameworks to uncover strengths and weaknesses of each framework.

**2 Study area and benchmark data**

The study area is situated in the North Eastern part of Spain (Figure 1a) and constitutes a smaller part of the lower Ebro River basin. The area is characterized by a semi-arid climate with yearly mean temperatures of around 15 °C and precipitation of approximately 400 mm/year (La Agencia Estatal de Meteorología: https://www.aemet.es/, last access: 31 May 2023). Precipitation mainly occurs in April–May and October–November, and summer is characterized by sporadic precipitation and high temperatures. The plains in the lower part of the sub-catchment are covered by cropland (Figure 1b) that is further

subdivided into four central irrigated districts, mainly with summer cereal, forage, and fruit trees, and surrounded by drylands with cereals and olive groves. Yearly crop and irrigation practice maps from Catalonia are available via the Geographic Information System for Agricultural Parcels (SIGPAC), provided by the *Department d'Agricultura, Ramaderi, Pesca i Almentació*. https://analisi.transparenciacatalunya.cat.

   The irrigation districts are of varying sizes and are linked to specific reservoirs connected, in turn, to a network of

canals that deliver water to the fields. The districts included in this study are: 1) The Urgell district (887.62 km$^2$) is the oldest irrigation district and largely applies biweekly flood irrigation, supplied by the Rialb (measured at canel C116) and Sant Llorenç de Montgai reservoirs (measured at canel C117) (Figure 1b). 2) The Algerri Balaguer district (70.79 km$^2$) is supplied with irrigation water from the Santa Ana reservoir (measured at pump E271) and mainly applies sprinkler and drip irrigation techniques, which can occur daily. 3) The Catalan Aragonese district (1161.52 km$^2$) irrigation water is supplied by the Barasona

(measured at canel C081) and Santa Ana reservoirs (measured at canel C101). The water is applied by different irrigation techniques; Flood irrigation (18%), drip irrigation (28%), and sprinkler irrigation (54%) (Dari et al., 2020). We assume that most irrigation water to the Northern part of the district is supplied by the Barasona Reservoir and the Southern part by the Santa Ana Reservoir (Figure 1b). Data on irrigation water use from each of the main reservoirs are collected by the Automatic Hydrologic Information System of the Ebro River basin (SAIH Ebro) (available at: http://www.saihebro.com). The data is

used as a benchmark for the period 2016 – 2017.

   Dari et al. (2020) estimated expected water losses due to irrigation efficiency based on literature and irrigation techniques adopted within each district that were applied to the benchmark irrigation volumes: Urgell (30% loss), Algerri Balaguer (10%), North and South Catalan Aragonese (15%). In the Urgell, irrigation still occurs through a traditional flood irrigation network, which explains the highest loss rate adopted for such districts. Algerri Balaguer is equipped with a modern

system in which drip irrigation is employed for fruit trees and sprinkler irrigation is adopted for herbaceous crops (Dari et al., 2021); hence, a lower loss rate (10%) has been adopted here. Finally, mixed techniques are employed over the Catalan and Aragonese districts (with sprinkler irrigation predominating), and thus an intermediate loss rate has been considered. For the Urgell district, data are missing from July 2016 to February 2017 and data are missing for the North Catalan Aragonese district for September and October.

## 3 Method and data

With this study, we aim to compare irrigation estimates from soil moisture- and ET-based approaches. The framework used herein aims to isolate a non-precipitation source of water (either soil moisture or ET) from satellite observations by subtracting a hydrological modeled rainfed baseline (Koch et al., 2020; Zaussinger et al., 2019). We present results based on four baseline models; one calibrated solely against rainfed ET, two calibrated against rainfed soil moisture with different adjustments (further described in section 3.1), and one model calibrated jointly against ET and soil moisture. The aim is to keep the framework as similar as possible for all four approaches. Figure 2 presents the overall approach for the four model calibrations and irrigation quantification steps and similarities and differences are highlighted. The differences will be further described in Section 3. We also include published irrigation estimates over the same study site obtained through an SM-based inversion framework forced with different soil moisture data sets as described in Dari et al. (2020, 2023)

### 3.1 Evapotranspiration and soil moisture data

Information on ET and soil moisture quantities and patterns are available from a variety of satellite remote sensing systems. In this study, we choose the MODIS 16 ET product (Mu et al., 2007, 2011) that among other products was found to provide valid ET estimates across several European catchments and to be influenced by irrigation (Dari et al., 2022b; Stisen et al., 2021). We also used a SMOS soil moisture dataset downscaled by the DISPATCH algorithm (Merlin et al., 2013), that have been evaluated and used within an SM-based inversion framework to quantify irrigation amounts and patterns within the study area (Dari et al., 2020, 2021).

The MODIS 16 ET product estimates ET at 500 m spatial resolution as an 8-day total flux [mm] of soil-canopy evaporation and plant transpiration. The estimates are vegetation-based as the surface energy balance is constrained by both meteorological reanalysis data and MODIS vegetation properties, albedo, and land cover (Cleugh et al., 2007) as input within the Penman-Monteith equation (Monteith, 1965). The ET fluxes were aggregated from 8-day to monthly estimates by multiplying the mean daily flux for each month by the number of days.

The SMOS product estimates soil moisture at a 35 – 50 km spatial resolution as a daily volumetric water storage [$m^3/m^3$] of the topsoil (Kerr et al., 2012). The estimates are based on the soil emissivity of microwaves that depend on the soil moisture content due to the large dielectric difference between dry soil and water (Kerr et al., 2001). The SMOS product has been downscaled by using soil evaporative efficiency (SEE) at 1 km resolution, estimated from MODIS NDVI and LST, to disaggregate the original SMOS pixel by redistributing the values according to SEE but maintaining the original volumetric water content by averaging the downscaled product to SMOS' native resolution (Merlin et al., 2013). Due to its low resolution, the original SMOS dataset cannot effectively capture the irrigation signal (Kumar et al., 2015), which was first introduced through the DISPATCH downscaling algorithm. The original SMOS DISPATCH near-surface soil moisture was used, and a root zone soil moisture estimate was derived by applying a recursive exponential filter equation proposed by Albergel et al.

(2008). To convert the product from volumetric water storage to water depth, we assumed a constant sensing depth of 5 cm (Kerr et al., 2001). The soil moisture products were used at a daily resolution to quantify irrigation although less robust than monthly because the aggregation of the daily soil moisture storage, compared to ET flux, proved to be very uncertain.


## 3.2 Hydrological models

The grid-based mesoscale Hydrological Model (mHM, Kumar et al., 2013; Samaniego et al., 2010; Thober et al., 2019) version 5.11.0 (Samaniego et al., 2021) was used to model rainfed ET and soil moisture baselines. mHM yields consistent spatial parameter distributions across scales using the multiscale parameter regionalization technique (Schweppe et al., 2022) that,
via nonlinear transfer functions, links parameter distributions at an intermediate model scale to a fine-scale variability of spatially distributed catchment attributes. Seamless model parameter distributions are connected to a low number of global parameters that allow for a simple and powerful calibration (Samaniego et al., 2021, 2017). For this study, the hydrological models were calibrated and executed at 500 m spatial resolution for the period 2016 -2017, using 1 km gridded meteorological forcing and 250 m morphological input data. Soil moisture for each model layer is calculated as the effective precipitation
minus the actual evaporation from open water bodies and infiltration to deeper layers or recharge to groundwater. The actual ET is deduced from a reduction of PET by the Feddes soil water stress factor (Feddes et al., 1976) and a root fraction distribution factor over the defined number of soil layers.

Precipitation data were acquired from ERA5-Land (Muñoz Sabater, 2019), the daily average air temperature was acquired from Eobs (Cornes et al., 2018), and potential ET was calculated by the Hargraves equation (Hargreaves and Samani,
1985) from Eobs daily average, minimum and maximum air temperature and downscaled to model resolution by monthly climatologies of leaf area index (retrieved from MODIS MCD15A2H.v006) (Demirel et al., 2018). Hargraves were chosen to estimate potential ET due to their simplicity and based on results from Dari et al. (2022) who found Hargraves to be a good approximation, compared to the more complex Penman-Monteith when estimating actual ET from irrigation. All meteorological forcing data were resampled to a spatial resolution of 1 km using a bilinear function. The DEM was acquired
from NASA's Shuttle Radar Topography Mission data (Jarvis et al., 2016), and soil texture was acquired from the SoilGrid$^{TM}$ database (ISRIC, 2020) for six horizons (layer thickness from top: 5 cm, 10 cm, 15 cm, 30 cm, 40 cm, and 1 m). All morphological data were resampled to a spatial resolution of 250 m using the mean function. Land use, classified as pervious, impervious, and forest, was acquired from CORINE land use and Copernicus Land monitoring imperviousness datasets from (https://land.copernicus.eu/pan-european) (© European Union, Copernicus Land Monitoring Service 2018, European
Environment Agency (EEA)) and was resampled to a spatial resolution of 250 m using mode function.

### 3.3 Calibration strategy

The calibration framework is designed to obtain hydrological models that simulate robust baselines of rainfed ET and soil moisture. The Optimization Software Toolkit – OSTRICH (Matott, 2017) includes a Pareto Archived Dynamically Dimensioned Search (PADDS) algorithm (Asadzadeh and Tolson, 2009) that was used to calibrate the four hydrological baseline models. An initial sensitivity analysis determined that ten parameters were needed to be included in the one model calibration solely against ET, and 14 parameters were needed in the three other model calibrations that were either partly or fully calibrated against soil moisture. The three calibrations with soil moisture as the target included the same 10 parameters as the ET calibration and an additional four parameters related to soil and root fraction characteristics.

The objective functions were used to target the magnitude and temporal dynamic of ET and soil moisture over rainfed cropland. The rainfed cropland used as calibration target area was mapped in two steps: first, separate temporal stability analyses (Vachaud et al., 1985) were performed on the MODIS 16 ET and SMOS DISPATCH products to map areas drier than average (rainfed cropland), and second, the two maps were compared and pixels appearing as rainfed cropland in both maps were included as calibration target areas. Results from the temporal stability analysis can be found in supplementary materials (Figure S1). Only year-round rainfed cropland was included in the calibration because some irrigation is known to occur in the irrigation districts in between the main irrigation season. First, the magnitude was targeted by minimizing the mean absolute error (MAE) over all rainfed cells for each timestep during the entire two-year calibration period, Eq. (1).

$$MAE = \frac{\sum_{i=1}^{n}|x_i - y_i|}{n},\tag{1}$$

$$r = \rho(x, y),\tag{2}$$

Where $x_i$ and $y_i$ represent observed and simulated values at cell $i$, and $n$ is the number of observations. MAE varies between an optimal value of 0 to positive infinity. Second, dynamics were targeted by maximizing Pearson's correlation coefficient (r) on mean monthly quantities, Eq. (2), where $x$ and $y$ denote observed and simulated values. Pearson's correlation coefficient varies between an optimal value of -1 to 1. To select the best parametrization from the Pareto front the solution with the lowest normalized sum, concerning best-performing solutions, of the objective functions was chosen.

For the soil moisture-based and joint soil moisture and ET-based approaches, the soil moisture model outputs were rescaled within each iteration of the calibration by first subtracting the modeled mean soil moisture content and then adding the satellite reference mean soil moisture content. This was done to account for the systematic differences in how satellites with varying sensing depths and the hydrological model with a fixed top layer depth may respond to precipitation (Brocca et al., 2013; Zaussinger et al., 2019). The rescaling also shifts the focus of the model calibration towards parameters controlling the soil moisture variation rather than the average content, this is further important for reasons discussed in section 4.2.

### 3.4 Estimation of irrigation within a baseline framework

The hypothesis is that soil moisture and ET residuals between a remote sensing model and a hydrological-modeled rainfed baseline can be used to quantify irrigation (Koch et al., 2020; Zaussinger et al., 2019). The hydrological model parameters are calibrated for rainfed cropland and the seamless parameter fields generated by the parameter regionalization technique in mHM allow for a meaningful parameter transfer, which enables the model to simulate a robust rainfed baseline for irrigated cropland.

When calculating irrigation volumes, both ET and soil moisture residuals must be considered as none of the two components alone can fully capture the irrigation input. Earlier studies using the same baseline framework have been focussing on the quantification of net irrigation from ET residuals, which is the irrigation water loss to the atmosphere (Koch et al., 2020; Kragh et al., 2023). The total irrigation amount cannot be quantified from ET residuals alone as some of the water is bound within the soil column. Moreover, some irrigation water potentially infiltrates to deeper soil layers, recharges to groundwater, or generates overland flow. This, however, is thought to be a minor part of the total irrigated water amount, demonstrated by Dari et al. (2020) from the SM-based inversion framework and therefore not considered in this framework. Zaussinger et al. (2019) used the soil water balance equation to quantify irrigation by assuming all terms to be equal when comparing soil moisture changes from a rainfed hydrological model with a satellite reference except for the irrigation input measured by the satellite system. The method underestimated the irrigation amounts which could be a consequence of ignoring the fact that the actual ET is enhanced under irrigated conditions , thus missing a part of the irrigation signal.

This study proposes an extension of the baseline framework to quantify irrigation as the sum of soil moisture and ET residuals by subtracting rainfed model baselines from satellite references, Eq. (3).

$$irrigation = \left(SM_{satellite\ reference} - SM_{rainfed\ baseline}\right) + \left(ET_{satellite\ reference} - ET_{rainfed\ baseline}\right) \tag{3}$$

Where SM and ET denote soil moisture water depth and ET flux in mm, respectively. For the ET component, monthly irrigation amounts are estimated by calculating the mean across grid-specific ET residuals for each district. For the soil moisture component, monthly irrigation amounts are estimated, by first calculating daily mean soil moisture for each district (days with less than 50% coverage were excluded), followed by calculated daily district-specific soil moisture residuals and then summing the mean daily soil moisture residuals to monthly estimates. The spatial distribution of the soil moisture component is calculated by summing daily to monthly residuals to extract irrigation patterns, which are then rescaled to the monthly estimates for each district. Soil moisture and ET processes are interconnected and summing both components in Eq.3 may cause a double counting of water when estimating irrigation, leading to an overestimation of irrigation. However, due to the conversion from near-surface to root-zone soil moisture and the spatial aggregation of soil moisture, described in more detail below, we have alleviated the effect of double counting. Moreover, with the available benchmark data, described in section 2, we have the possibility to evaluate the estimated irrigation amounts which allows us to rule out a substantial deterioration of the estimated irrigation by double counting.

In theory, the SMOS DISPATCH near-surface soil moisture residuals capture most of the irrigated water use as most of the water will pass the soil surface before returning to the atmosphere. However, near-surface soil moisture is difficult to

represent via hydrological modeling because satellite observations provide simply a 'snapshot' of the hydrological state, which could have been measured during or just after a rainfall event. In the case of a satellite overpass just after a rainfall event, most of the soil water will be present in the near-surface, and during the following days, it will separate into deeper infiltration or evapotranspiration. Similarly, when a hydrological model simulates soil moisture during a rainfall event, the model will distribute the input between all soil layers and outgoing fluxes within a single timestep (one day in our study), which makes it challenging to directly compare model estimates with satellite observations.

We addressed this limitation by converting the SMOS DISPATCH near-surface soil moisture observation to a root zone estimate as proposed by Albergel et al. (2008) which is an attempt to account for the temporal processes that affect the distribution of soil moisture. Consequently, this will make the satellite observation more comparable with the hydrological model. During this conversion, we have removed around 40% of the summed near-surface soil moisture increases. Since evaporative losses from the near-surface soil layer are being removed from the SMOS DISPATCH dataset, we additionally need to consider ET contributions in Eq 3. Further, due to the spatial aggregation of daily mean soil moisture residuals at the district scale, the soil moisture component will represent a more conservative estimate of the applied irrigation. This is because the districts are not uniformly irrigated at the same time, and at the grid level, daily soil moisture residuals can be both negative and positive. Also considering negative soil moisture residuals in the calculation of the soil moisture component in Eq.3 captures to some degree the interconnection between soil moisture and evapotranspiration  Due to the substantial removal of near-surface soil moisture input (40%) from the conversion to root zone soil moisture and the spatial district aggregation, we do not expect that our irrigation estimates will be severely affected by double counting of water through the joint analysis of ET and soil moisture residuals. We do acknowledge that if this method were to be used in an area with known over-irrigation issues, drainage, and overland flow terms would need to be added to Eq (3) to fully represent irrigation water use.

As a stand-alone approach, to quantify irrigation volumes solely from soil moisture residuals, a hydrological baseline model calibrated on near-surface soil moisture was also tested. The hypothesis for using near-surface soil moisture instead of root-zone soil moisture is that by preserving the original raw observations that potentially contain all water entering the soil, we can calculate the irrigation volume from soil moisture residuals without having to consider the ET component.

## 3.5 Other irrigation estimates and frameworks

Three irrigation estimates from an SM-based inversion framework (Brocca et al., 2018) are included, which use soil moisture observations from DISPATCH downscaled observations from SMOS and SMAP satellites (SMOS_if and SMAP_if) (Dari et al., 2020) and first-order radiative transfer modelling (Quast et al., 2019, 2023) of Sentinel-1 backscatter (S1RT_if) (Dari et al., 2023) (Table 1). Within both studies (Dari et al., 2021, 2023), the soil moisture observations were referred to as surface soil moisture. In this study, we refer to all soil moisture observations filtered with the recursive exponential filter equation proposed by Albergel et al. (2008) as root zone soil moisture and original observations as near-surface soil moisture. These studies took place at the same study site and are a display of already published datasets.

Here, is a short description of the SM-based inversion framework as a foundation to understand the irrigation estimates, and we refer to Dari et al. (2020, 2023) for in-depth method descriptions. First, the SM-based inversion framework builds on rearranging the soil water balance equation to enable backward calculation of total water input (precipitation plus irrigation) from soil moisture observations and isolate the irrigation signal from the estimated total water input by subtraction of precipitation. The prevailing part of the total water input in the SM-based inversion framework is expressed by changes in soil moisture storage and ET flux, which are the same two components describing the irrigation-induced residuals between reference and baseline models within the baseline framework.

In both Dari et al. (2020, 2023) calculations of water input from changes in soil moisture storage and drainage are similar but differ regarding the ET flux term. Dari et al. (2020) follow guidelines provided by the FAO paper 56 (Allen et al., 1998) and use crop coefficients that incorporate the influence of soil moisture stress on transpiration. Dari et al. (2023) follow a soil moisture limiting approach and combine the soil moisture index and a bias correction factor to estimate ET from PET. To calibrate the SM-based inversion framework, Dari et al. (2020) calibrated the soil parameters of the three algorithms against precipitation at rainfed cropland and transferred the median of spatially distributed parameters to irrigated cropland. Conversely, Dari et al. (2023) implemented an iterative strategy to obtain spatially distributed soil parameter values calibrated against rainfall and a fixed value for the ET adjusting factor calibrated by considering, as a benchmark, rainfall plus irrigation over selected sites.

## 4 Results and Discussion

### 4.1. Calibration results

The analysis of soil moisture and ET approaches within the rainfed baseline framework builds upon results from four approaches: the ET approach by calibration against MOD16 ET (ET_bf), two representing the soil moisture approach by calibration against SMOS DISPATCH near-surface (NS-SM_bf) and root-zone soil moisture (RZ-SM_bf), and one representing a joint root-zone soil moisture and ET approach (joint_bf) calibrated against both references (Table 1). All models were separately calibrated for rainfed cropland conditions. Based on the four Pareto fronts, the four solutions with the lowest normalized sum were selected (Table 2).

**Table 1: Overview of investigated frameworks and approaches**

| abbreviation | framework / approach | calibration target |
|---|---|---|
| RZ_SM_bf | baseline framework / soil moisture | root zone soil moisture (from SMOS DISPATCH) |
| joint_bf | baseline framework / soil moisture and ET | root zone soil moisture (from SMOS DISPATCH) and MOD16 ET |
| ET_bf | baseline framework / ET | MOD16 ET |
| NS_SM_bf | baseline framework / soil moisture | original SMOS DISPATCH near-surface soil moisture |
| SMOS_if | inversion framework / soil moisture | root zone soil moisture (from SMOS DISPATCH) |

| | | |
|---|---|---|
| SMAP_if | inversion framework / soil moisture | root zone soil moisture (from SMAP DISPATCH) |
| S1RT_if | inversion framework / soil moisture | root zone soil moisture (from Sentinel-1 RT) |

315

**Table 2: Calibration results for four baseline frameworks. Statistics are calculated over rainfed cropland for the period 2016 – 2017.**

| calibration results | RZ-SM_bf | joint_bf | ET_bf | NS-SM_bf |
|---|---|---|---|---|
| MAE | 2.8 mm soil moisture/day | 2.4 mm soil moisture/day<br>8.1 mm ET/month | 7.1 mm ET/month | 2.8 mm soil moisture/day |
| Pearson correlation | 0.84, soil moisture | 0.82, soil moisture<br>0.82, ET | 0.84, ET | 0.76, soil moisture |

The joint_bf baseline has a similar ET performance as ET_bf but has better soil moisture performance than the NS-SM_bf and RZ-SM_bf baselines because it benefits from targeting both the soil moisture and ET references that enable the model to better simulate soil moisture dynamics. The NS-SM_bf baseline has the lowest Pearson correlation because the NS-SM reference has more day-to-day variation that cannot be simulated by the hydrological model compared to the RZ-SM reference, which has a smoother trajectory.

In general, all baseline models exhibit the poorest performance during winter and early spring, and it is known from other irrigation studies that irrigation estimation is more uncertain during rainy periods, which makes the separation between precipitation and irrigation-induced changes more difficult (Brocca et al., 2018; Dari et al., 2020; Jalilvand et al., 2019; Koch et al., 2020). Time series from the model calibrations of rainfed cropland can be found in supplementary materials (Figure S2-S6) together with bias measures of mean error (ME) and the standard derivation of rainfed residuals (Table S1).

**4.2 Soil moisture and ET approaches within a rainfed baseline framework**

The four approaches produce irrigation estimates that overall match the benchmark (Figure 3), In 2016, irrigation estimates are linked with precipitation by showing low irrigation supply when precipitation is high and vice versa during summer. In 2017 the irrigation estimates steadily increased over a prolonged period due to the uniform precipitation input throughout the year. Thus, the timing and intensity of the irrigation estimates seem to be affected more by precipitation variability than the benchmark and show that large amounts of precipitation counteract the irrigation requirements. This coupling between irrigation estimates and precipitation gives confidence to the applied framework's ability to create robust baselines able to separate precipitation-related changes from a remote sensing reference.

The joint_bf estimate (stacked graph, Figure 4) is characterized by the soil moisture storage increase in 2016 due to irrigation, followed by storage decrease as ET rapidly depletes the root zone soil moisture storage. In 2017, soil moisture

storage and ET simultaneously increased and decreased due to the meteorological conditions that did not support the same rapid depletion of the root zone as in 2016. The joint_bf is overall close to the benchmark and captures the different aspects of the hydrological cycle that cannot be captured by a stand-alone analysis of soil moisture or ET.

The NS-SM_bf estimate (black graph, Figure 4) matches the average timing and intensity of the joint approach in 2016 as it captures both soil moisture storage and ET flux changes induced by irrigation. In 2017 the irrigation estimate is a little lower than the joint approach, possibly because of the precipitation input that causes an overestimation of the baseline, thus underestimating irrigation during the irrigation season and estimating almost no irrigation in-between irrigation seasons. To address this, further work is needed on the calibration of near-surface rather than root-zone soil moisture. However, the NS-SM_bf estimate points towards a methodology to quantify irrigation solely based on soil moisture storage changes.

The distribution of irrigation water is more complex than direct allocation from the main reservoirs to the fields (benchmark data) as smaller reservoirs also exist within the districts, which means that there might not be temporal consistency between the allocation and field practice as water can be stored for days or months in between irrigation seasons. Still, the benchmark provides an upper limit when evaluating irrigation estimates, which could be done on accumulated benchmark volumes over longer periods to account for the delay between the allocation and timing of irrigation.

In the Algerri Balaguer district (Figure 3), the spring irrigation increases above the benchmark in 2016 and 2017, which also can be seen in the Urgell, Algerri Balaguer, and South Catalan Aragonese districts, during autumn in 2017 when precipitation was low. These results suggest that water stored during the current season or from last season is used for irrigation.. Irrigation estimates from radar soil moisture also suggest that water storage within the districts could play a significant role in this context (Dari et al., 2023).

In 2016, the irrigation estimates were lower than the benchmark (Figure 3) in the Urgell, North, and South Catalan Aragonese districts because the rainfed baselines, linked to the spring precipitation (183 mm), explain most of the observed flux and storage changes. The benchmarks from 2016 and 2017 have very similar timing and intensity, although the precipitation patterns are very different. In 2016 the precipitation input was high during winter and spring, low during summer, and high during autumn, whereas 2017 lacks the expected seasonality with 25 mm/month precipitation on average besides a wet March. This could suggest that the reservoirs might work according to a relatively fixed schedule or that not all of the benchmark water might be allocated to irrigation but simply is balancing the reservoir water level.

The baseline framework offers a solution to both separately or jointly utilize soil moisture and ET in calibrations to extract consistent and robust rainfed baselines. The calibration targets were similar for both soil moisture and ET to keep the approaches inter-comparable although soil moisture is more challenging to calibrate, compared to ET. This challenge arises because the hydrological model and the satellite-derived products react differently to precipitation as it implies that the modeled baseline must be rescaled to the reference mean content, which limits the calibration of sensitive soil parameters to only satisfy the fit to the seasonal amplitude.

The SMOS DISPATCH product is characterized by low mean soil moisture content and high seasonal amplitude, which poses an issue due to a positive relationship between mean content and amplitude within the mHM model. This pushes

the baseline calibrated for root zone soil moisture (RZ-SM_bf) to simulate a mean soil moisture content that is 10 mm higher than SMOS DISPATCH to fit the amplitude, whereas the baseline solely calibrated against ET (ET_bf) simulates a mean soil moisture content only 2 mm higher than SMOS DISPATCH but underestimates the amplitude. This creates a seasonal bias that underestimates irrigation estimates during summer and overestimates during winter (Figure 4). On the contrary, the ET baselines are more comparable between the three approaches because the relationship between the field capacity and wilting is scaled according to the modeled amplitude, which yields similar soil-water stress factors used to estimate actual ET.

The RZ-SM_bf and ET_bf results represent the highest and lowest irrigation estimates during the main irrigation season respectively (Figure 4), mainly due to large differences in the soil moisture-based component of the irrigation estimation. This is supported by comparing mean monthly coefficients of variation (CV) of the ET (CV = 0.14) and soil moisture (CV = 0.45) components between ET_bf, RZ-SM_bf, and joint_bf. This shows that the soil moisture baselines are difficult to estimate without a soil moisture calibration target as compared to the ET baselines, which can be estimated fairly accurately only with a soil moisture calibration target. This also calls for further work on how to improve soil moisture calibration by targeting attributes such as field capacity, wilting point, amplitude, or other soil moisture characteristics as proposed by Araki et al. (2022). Only one combination of soil moisture and ET references was used in a joint calibration, and further testing by ensemble analysis is required to establish guidelines on how to combine different references and fully understand what specific attributes from soil moisture and ET that need to be targeted to extract valid baselines.

## 4.3 Irrigation frameworks, their strengths, and weaknesses

The comparison of frameworks for irrigation quantification is based on our four baseline framework estimates (ET_bf, RZ-SM_bf, NS-SM_bf, and joint_bf) and three SM-based inversion framework estimates (SMOS_if, SMAP_if, and S1RT_if).

Mean irrigation estimates for the entire irrigated cropland (Figure 5) show that all estimates capture the irrigation seasonality that is expected to peak during summer. The dynamics of the four baseline framework estimates are similar because they are based on the same references and meteorological forcing, likewise SMOS_if and SMAP_if. These six estimates use similar DISPATCH downscaled references and meteorological forcing and therefore produce generally similar spatiotemporal irrigation patterns. The S1RT_if estimate seems to vary more with precipitation than SMOS_if and SMAP_if (Figure 5), which is probably an effect of the spatially distributed soil parameters that are fitted for each pixel to the precipitation and benchmark input. As mentioned in section 4.1, the S1RT_if estimates suggest the use of stored irrigation water within the districts around May, seen as a peak each May in Figure 5, but based on the reoccurring irrigation peaks each May, within each district, this could also be explained by the reoccurring influence of vegetation that is known to influence radar signals (Meyer et al., 2022). However, the temporal correlation is high between all seven estimates, ranging between 0.69 and 0.97. Results from a temporal and spatial correlation analysis are presented as a correlogram in Figure 6.

One of the apparent differences among the estimates is a weaker response between precipitation and irrigation amounts in the SM-based inversion framework compared to the baseline framework as pointed out in section 4.2, which for

SMOS_if and SMAP_if could be a consequence of simplifying the ET calculation by implementing a water-based limitation approach, or maybe relate to the PET product used as input. Dari et al. (2020) analyzed the contribution from soil moisture and ET components of the SMAP_if estimate, which showed a larger ET contribution during spring (when precipitation and soil moisture content are high) than what is estimated by the baseline frameworks in this study (Figure 4). The main purpose of using a rainfed hydrological model is to estimate how much of the observed ET can be explained by the precipitation input and thereby is embedded within the baseline, making sure that when larger amounts of precipitation input are available the irrigation estimates will be low if the baseline is well calibrated. PET estimates can vary substantially between readily available remote sensing datasets or estimates from meteorological data, which ultimately can have a profound impact on the irrigation components if not accounted for in the calibration (Kragh et al., 2023).

Overall, there is a tendency to underestimate irrigation (Table 3), either suggesting that the rainfed baselines are too high, total input estimates are too low, or that the irrigation loss might be even larger. Underestimation is however expected to some degree for the SM-based inversion framework estimates as each daily satellite overpass does not provide full coverage. The RZ-SM_bf and joint_bf estimates differ the least from the benchmark, in total underestimating irrigation with 6 and 2 mm yearly and overestimating with 47 and 37 mm on a seasonal basis (Table 3). The S1RT_if estimate differ the most from the benchmark because it estimate much lower irrigation in between irrigation seasons, and in addition, the S1RT_if is also underestimating during the main irrigation season. Yearly S1RT_if differ by -375 mm, but both show better performance on a seasonal basis by -285 mm (Table 3). As mentioned in section 4.2, a question may arise whether the benchmark in-between irrigation seasons does represent water used for irrigation. The ET_bf, NS-SM_bf, SMOS_if, and SMAP_if estimates differ yearly by -154, -143, -112, and -135 mm, respectively. ET_bf and SMOS_if perform slightly worse on a seasonal basis by -208 and -144 mm, respectively (Table 3), because they tend to overestimate irrigation in between irrigation seasons, which compensates for the underestimation during the main irrigation season. SMAP_if performs the same on a seasonal basis by -138 mm. NS-SM_bf performs better on a seasonal basis by -76 mm because it behaves reasonably well during the main irrigation on a more similar level to the RZ-SM_bf and joint_bf approaches.

**Table 3: Yearly and seasonal (April - October) irrigation estimates (mm) for different approaches and frameworks. The red and blue values show the under or overestimation of irrigation compared to a weighted area average benchmark in the last column. \* benchmark data is without observations from September to December 2017 for comparison with SMOS_if and SMAP_if estimates.**

| estimates | | RZ-SM_bf | joint_bf | ET_bf | NS-SM_bf | SMOS_if | SMAP_if | S1RT_if | Benchmark |
|---|---|---|---|---|---|---|---|---|---|
| 2016 | year | 553 (+25) | 535 (+10) | 477 (-46) | 465 (-49) | 495 (-37) | 436 (-96) | 375 (-157) | 532 |
| 2017 | year | 576 (-19) | 576 (-8) | 467 (-108) | 455 (-94) | 409 (-75) | 445 (-39) | 386 (-217) | 603 484* |
| sum | year | 1128 (+6) | 1111 (+2) | 944 (-154) | 920 (-143) | 905 (-112) | 881 (-135) | 760 (-375) | 1135 1016* |
| | | | | | | | | | |
| 2016 | season | 460 (+37) | 437 (+16) | 328 (-94) | 396 (-21) | 367 (-56) | 323 (-100) | 318 (-105) | 423 |
| 2017 | season | 504 (+10) | 502 (+21) | 366 (-114) | 402 (-55) | 314 (-88) | 364 (-38) | 320 (-180) | 499 401* |
| sum | season | 963 (+47) | 939 (+37) | 694 (-208) | 799 (-76) | 681 (-144) | 687 (-138) | 638 (-285) | 922 824* |

When comparing irrigation patterns from the baseline framework with aerial photos the irrigation patterns match the
435 surface greenness (Figure 1 and Figure 7) which is incorporated through the MODIS vegetation products by the ET reference, DISPATCH downscaling, and model vegetation input. The baseline framework estimates are very similar as they are produced by the same conceptual model (even though the calibration parameters are different), meaning that most of the spatial distribution of the irrigation estimates is locked within the model inputs.

The similarities between the SMOS_if, SMAP_if, and the baseline framework estimates are high, which is not that
surprising as they all use the same SMOS DISPATCH reference or a similar SMAP DISPATCH reference. The spatial distribution of S1RT_if irrigation is different from the other estimates. However, by closer comparison, there are similar large-scale features between all seven estimates which gives confidence to the S1RT_if estimate. An initial analysis within the study area of the DISPATCH downscaled products and the soil moisture data sets they originate from showed that the irrigation signal was forced into the soil moisture reference by downscaling, which is another reason why the Sentinel-1 RT product is
an interesting reference to include and further investigate.

The uncertainty introduced by using a daily soil moisture reference is not only an issue within the baseline framework but is something that also needs to be handled within the SM-based inversion framework. The SMOS_if and SMAP_if estimates are temporally aggregated from daily to 5-day estimates, which in addition to the lower spatial resolution (1 km) than the baseline estimates, will result in a smoother irrigation pattern when compared to the baseline framework estimates
(Figure 7). For the S1RT_if estimate, aggregation of daily irrigation is less of a limiting factor because the soil parameters for each irrigated cell are calibrated against precipitation and benchmark data, thus accounting for fine-scale soil moisture heterogeneity, which may lead to an over-parametrization.

The main advantage of the baseline framework is the consistency between the soil moisture and ET baselines making it unlikely that one component will control the calibration. For example, if the PET estimate used in the SM-based inversion
framework is too high, the contribution from the soil moisture and drainage components will need to be very low to correctly simulate the precipitation input, since ET will dominate. Thus, adjusting the calibration parameters will mostly be controlled by the ET component (Dari et al., 2022b). However, calibration of a hydrological model is not trivial as it includes many parameters to correctly split the precipitation input into the observed spatiotemporal patterns of rainfed soil moisture and ET, compared to the SM-based inversion framework that needs fewer parameters and uses the soil moisture balance equation to
directly convert the observed changes into a total of precipitation and irrigation.

The soil moisture component within the SM-based inversion framework is easier to handle as changes are directly converted to a volume based on calibrated soil parameters that contain information about soil layer depth and porosity. This eliminates the main issue within the baseline framework that arises from the mismatch between the unknown remote sensing depth of investigation and the static topsoil layer depth of a conceptual model (López et al., 2017; Reichle and Koster, 2004),
which makes it difficult to fully calibrate all sensitive soil parameters when rescaling is needed.

The ET component on the other hand is easier to handle within the baseline framework because the term can be adjusted to rainfed conditions through calibration, which is not a possibility within the SM-based inversion framework used

in Dari et al. (2020) as the term does not contain any calibration parameters causing the ET component to dominate how the total input is separated between the terms based on the PET estimate used. In Dari et al. (2023) the ET component is calibrated through adjustment of a bias correction factor to account for PET uncertainties, but the calibration target changes from rainfed to irrigated conditions. Nevertheless, this circumstance does not represent a limiting factor for the algorithm applicability, as it can be also implemented where reference irrigation data for calibration is not available by accepting a higher degree of uncertainty.

### 4.4 Influence of uncertainty

Based on the structure of the frameworks and their calibration strategies, the influence of various uncertainty sources can have different impacts on the irrigation estimates. This section aims to provide a brief overview of the main uncertainty sources and how much they impact the considered frameworks (Table 4).

Precipitation uncertainties have an indirect effect on the baseline framework which can potentially be mitigated by the calibration of soil moisture and ET baselines in the hydrological model (Kragh et al., 2023). In the soil moisture-based inversion framework, the influence is more direct as precipitation is used both as a calibration target and reference for quantifying irrigation water input as a residual (Dari et al., 2020).

ET uncertainties mainly become very apparent due to large differences across different remote sensing-based ET retrievals. The baseline framework addresses this uncertainty through model calibration to compensate for satellite biases (Kragh et al., 2023), whereas the soil moisture-based inversion framework can compensate by introducing a correction factor (Dari et al., 2023).

Soil moisture uncertainties mainly relate to the unknown sensing depth of the satellite systems. The soil moisture-based inversion framework addresses this uncertainty through calibration of a model parameter representing the water capacity of the soil layer (Brocca et al., 2018), where the baseline framework applies rescaling of model estimates of soil moisture to account for differences between model and satellite responses to precipitation. Overall, comparison of model estimates and satellite observations of soil moisture remains a challenge (López et al., 2017), which is avoided in the soil moisture-based inversion framework by directly converting observed soil moisture changes into irrigation. Overall, a common source of uncertainty related to soil moisture concerns its spatiotemporal resolution, which should match the spatiotemporal dynamics of irrigation (i.e., the spatial extent and the frequency) to catch the irrigation signal (Dari et al., 2022a; Zappa et al., 2022).

The uncertainties of the spatial resolution mainly relate to how well it allows the model domain to be classified into rainfed and irrigated cropland. The baseline framework is very dependent on a good classification of rainfed and irrigated cropland to be used in the calibration (Koch et al., 2020), whereas the soil moisture-based inversion framework is less sensitive since it can use both rainfed and irrigated cropland in the calibration (Dari et al., 2020, 2023). The uncertainties originating from the temporal resolution relate more to the calibration strategy and framework. The baseline framework works better with

more robust mean monthly observations compared to uncertain daily observations, whereas the soil moisture-based inversion framework requires a higher temporal resolution for the calibration.

The uncertainties of model parameters relate to how they are defined and spatially distributed. The baseline model has well-described physically based parameters that are distributed based on spatially distributed catchment characteristics using the multiscale parameter regionalization framework (Schweppe et al., 2022), whereas the soil moisture-based inversion framework applies coefficients describing processes either homogeneously or spatially distributed by extensive calibration of pixel-based coefficients (Dari et al., 2023). Dari et al. (2020) found that parameter uncertainty was low for the soil moisture-based inversion framework within a restricted region with homogeneous soil texture.

**Table 4: Overview of how different uncertainty sources affect the two frameworks. The +, ++, +++ symbology represents to what degree the framework may be influenced by an uncertainty source, where (+) is not much influence and (+++) is more influenced.**

| framework | precipitation | ET | soil moisture | resolution | parameters |
|---|---|---|---|---|---|
| soil moisture inversion | ++ | +++ | ++ | +++ | + |
| baseline | + | ++ | +++ | +++ | + |

## 6 Conclusion

This study aimed to compare the different state-of-the-art approaches and frameworks to quantify irrigation at a regional scale. To assess the strengths and limitations of irrigation estimates, we compared four separate and joint calibrations against soil moisture and ET references within a common baseline framework. Additionally, we compared three irrigation estimates derived from the SM-based inversion framework.

This study underlines the advantage of considering both soil moisture and ET residuals in a joint approach, by estimating irrigation during the main irrigation season (April - October) with an error of 37 mm to the benchmark (922 mm) over two years from 2016-2017. Through correlation analysis, we found that the baseline and SM-based inversion frameworks were able to produce similar spatial and temporal irrigation patterns with a correlation between 0.51 – 0.64 and 0.69 – 0.97, respectively, when using the same reference. The spatial correlation with irrigation estimates from the S1-RT_if was lower compared to the other estimates. However, the Sentinel-1 RT product used was the only independent soil moisture product in the inter-comparison and did produce temporal irrigation dynamics that correlated well with all other estimates (0.69 – 0.86), making it an interesting product to investigate further. The study also highlighted the importance of calibration strategies being tailored to both soil moisture and ET targets to estimate the right magnitudes of contribution from soil moisture and ET changes induced by irrigation to ultimately form more robust estimates. Also, the baseline framework was able to account for precipitation patterns through the rainfed baselines, which resulted in a more plausible temporal dynamic of the irrigation estimates than what was expected from the benchmark observations. This is an illustrative example of how we can gain knowledge about the hydrological system through hydrological models.

We found that uncertainty from daily soil moisture observations must be accounted for to quantify irrigation, either through the calibration or aggregation of estimates. We also found that the near-surface soil moisture approach could have the potential to estimate irrigation solely from soil moisture residuals, but further work on model calibration of near-surface soil moisture is needed.

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

**Data availability**

Irrigation estimates from the SM-based inversion framework using Sentinel-1 RT soil moisture (Dari et al., 2023) are freely available from https://doi.org/10.5281/zenodo.7341284. The four irrigation estimates from the baseline framework are available upon personal request (sjk@geus.dk).

**Author contribution**

SJK, JK, and JD designed the study and SJK carried it out in close consultation with JK and JD. SJK prepared the manuscript and figures in close consultation with JK and JD. All authors discussed results throughout the study period and provided critical feedback to the manuscript drafts and approved the final version of the manuscript.

**Competing interests**

The authors declare that they have no conflict of interest.

**Acknowledgments**

The authors wish to thank María José Escorihuela from isardSAT (mj.escorihuela@isardSAT.cat) for providing DISPATCH downscaled SMOS soil moisture data and feedback on the manuscript and the contribution by the ESA project "4DMed-Hydrology" (contract n. 4000136272/21/I-EF). This study was funded by the Independent Research Fund Denmark, project
number: 0164-00003B.


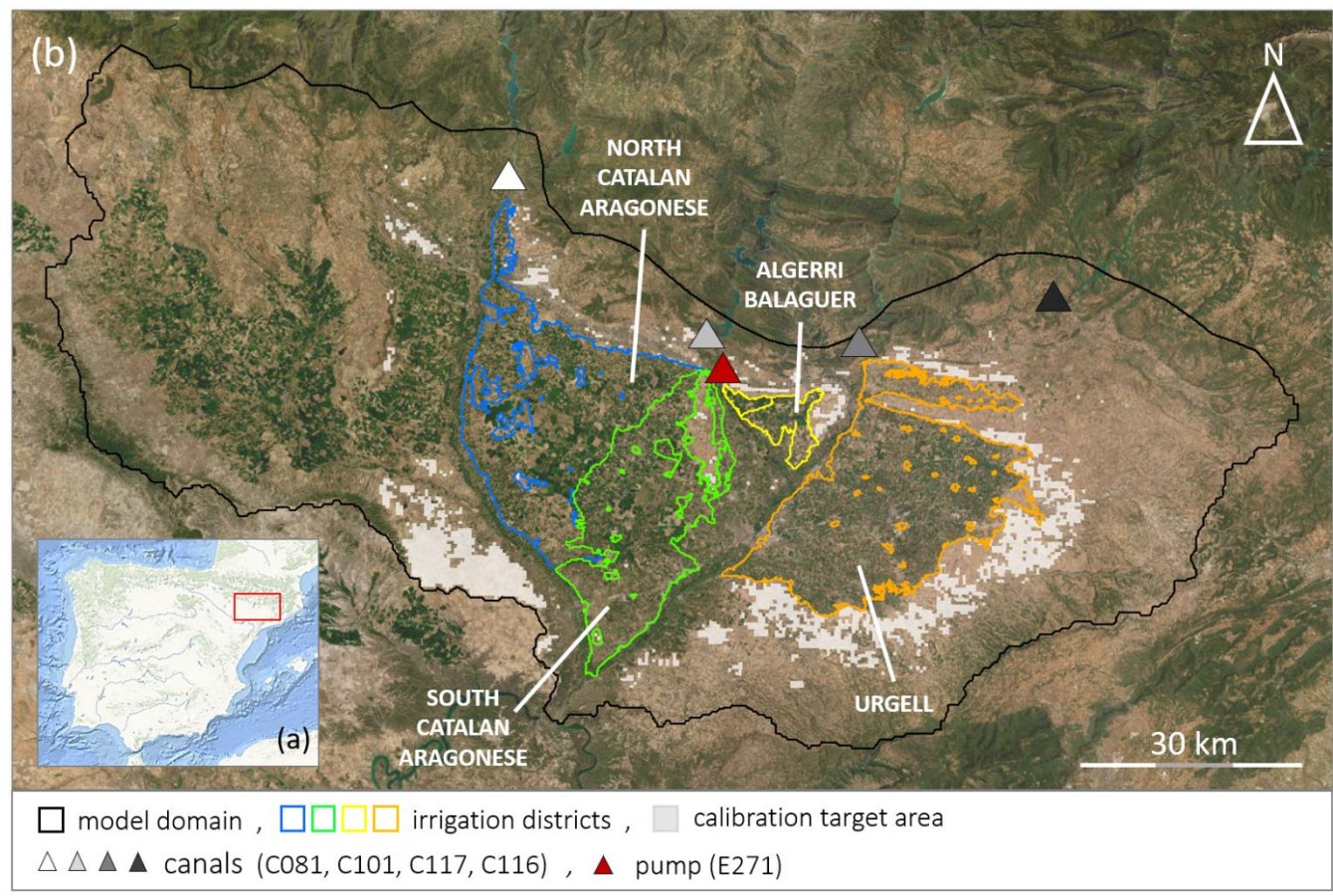


**Figure 1: Maps over the study area. (a) Map over the Iberian Peninsula (Source: Esri, Garmin, GEBCO, NOAA NGDC, and other contributors), the red square marks the extent of (b). (b) outline of the model domain, the four irrigation districts, canals, and pumps from where the benchmark was required. The light gray shaded area is rainfed cropland used for calibration. (Source: Esri, DigitalGlobe, GeoEye, Earthstar Geographics, CNES/Airbus DS, USDA, USGS, AeroGRID, IGN, and the GIS User Community)**




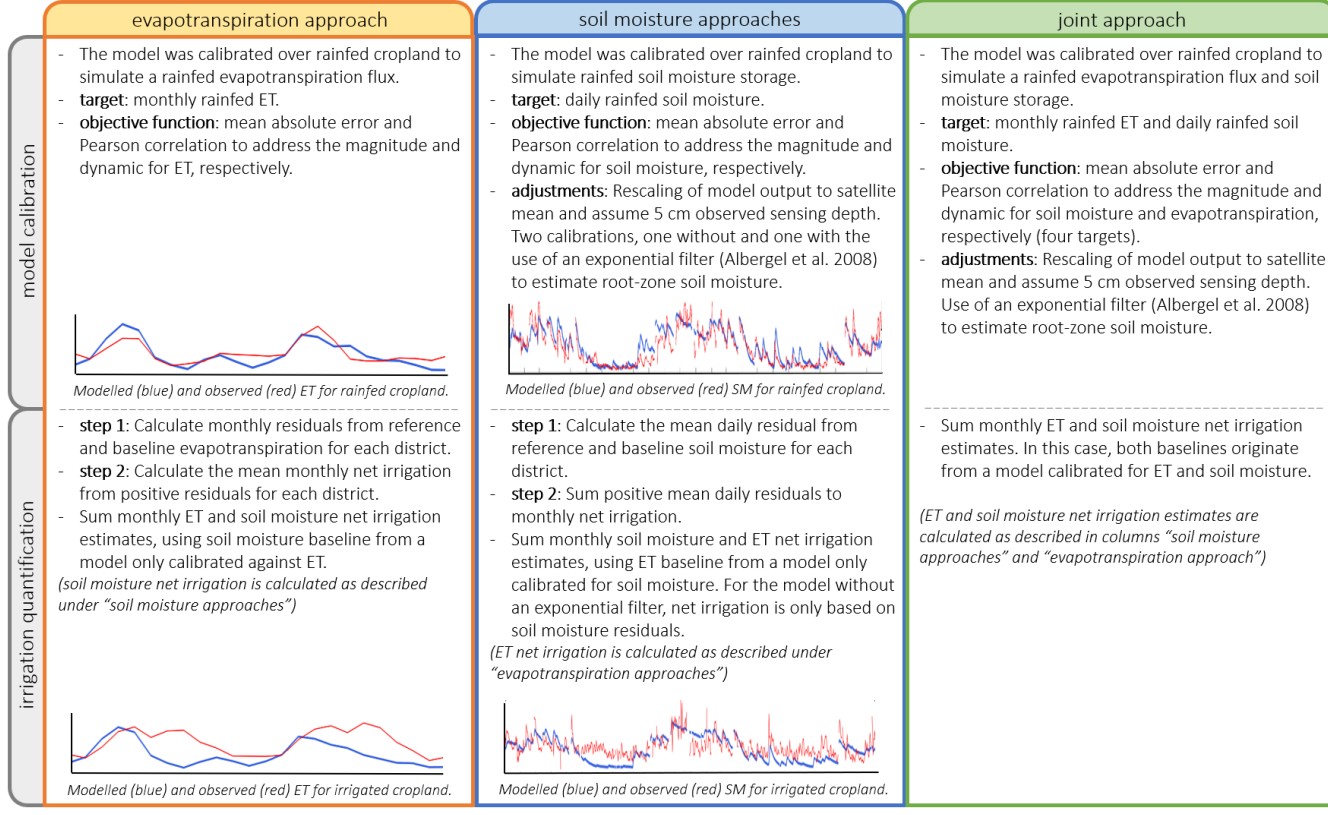

Figure 2: The scheme gives an overview of the similarities and differences between the four hydrological baseline models.



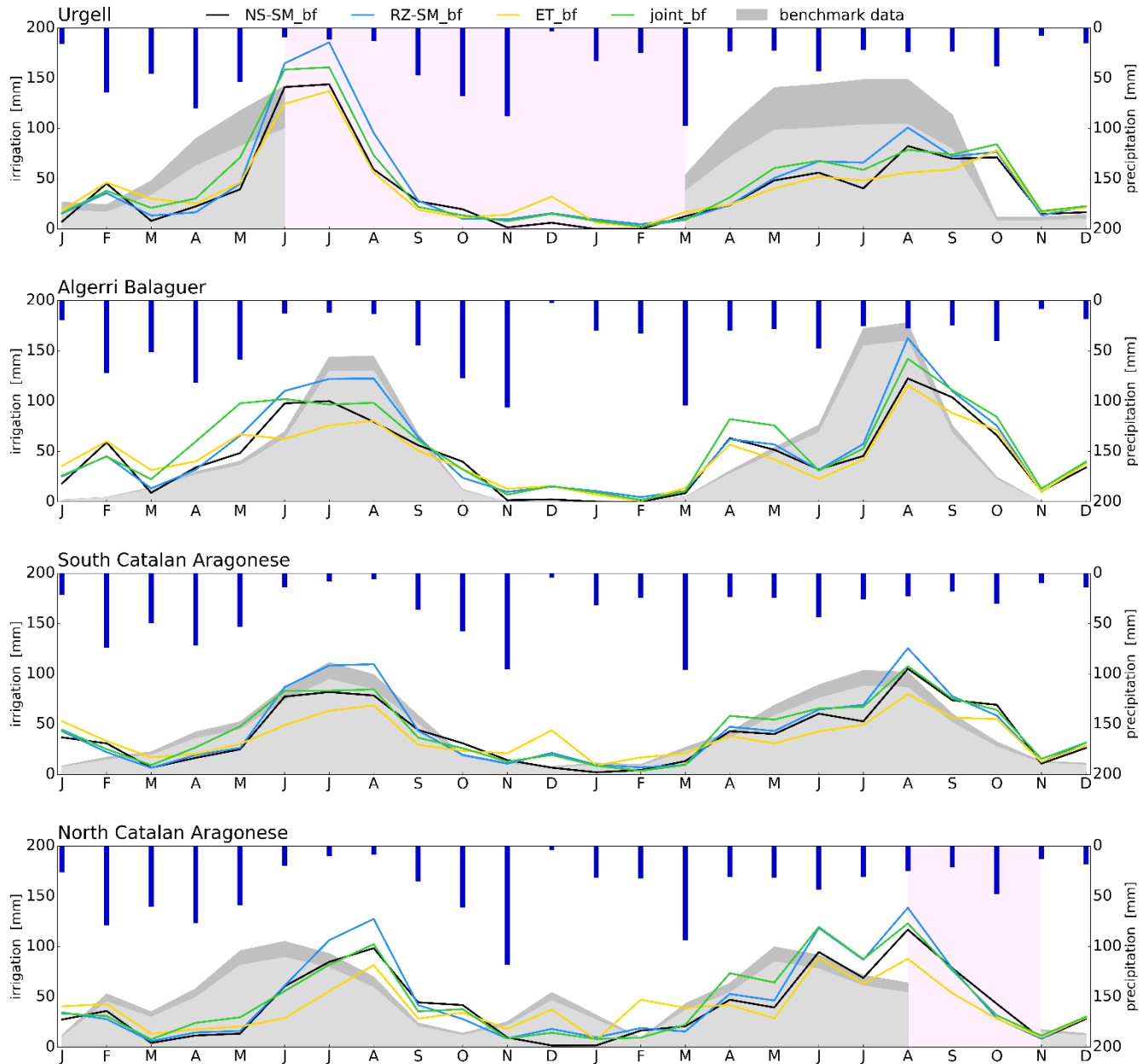

**Figure 3: Irrigation estimates from the four approaches within the baseline framework. Estimates are shown for each of the four irrigation districts for the period 2016 – 2017 and compared with the benchmark. The gray shaded area shows the benchmark with and without irrigation water losses due to irrigation efficiency. Dark blue bars show precipitation amounts. Light red areas mark missing benchmark observations.**

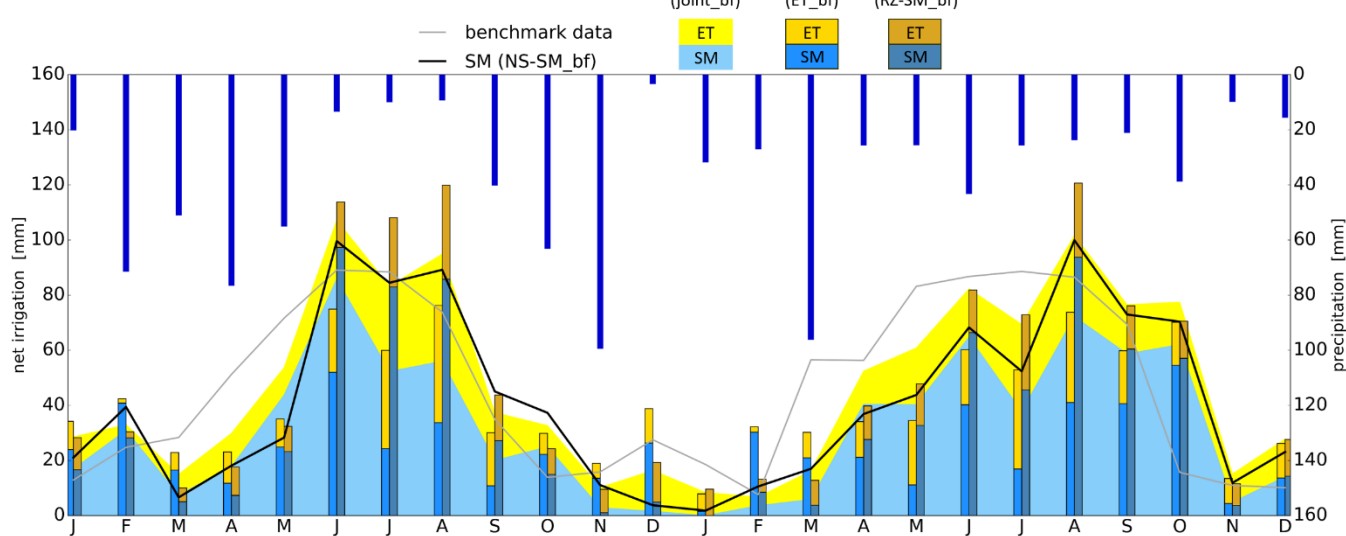


**Figure 4: Mean area irrigation estimates from the four approaches within the baseline framework. Estimates are shown for all four irrigation districts for the period 2016 – 2017 and separate the estimates between soil moisture (SM) and evapotranspiration (ET) contributions. The gray line represents a weighted area average benchmark data with losses due to irrigation efficiency. Dark blue bars show precipitation amounts.**


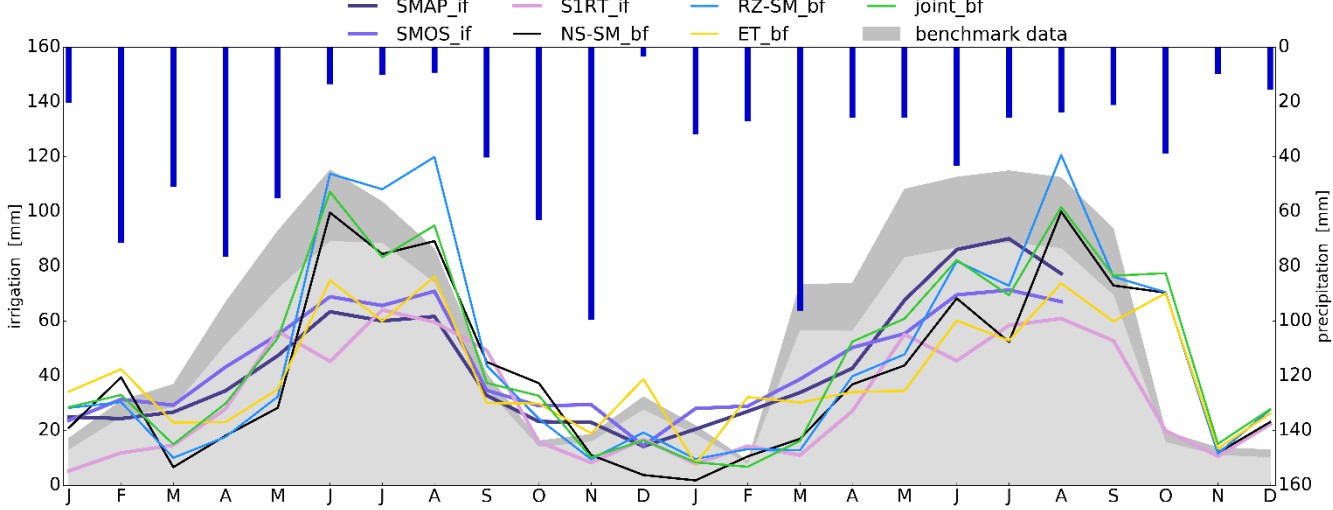

**Figure 5: Mean area irrigation estimates from the four baselines and three SM-based inversion frameworks. Estimates are shown for all four irrigation districts for the period 2016 – 2017. The gray shaded area shows a weighted area average benchmark with and without irrigation water losses due to irrigation efficiency. Dark blue bars show precipitation amounts.**


|  | RZ-SM_bf | ET_bf | joint_bf | NS-SM_bf | SMOS_if | SMAP_if | S1RT_if |
|---|---|---|---|---|---|---|---|
| RZ-SM_bf |  | 0.94 | 0.96 | 0.97 | 0.86 | 0.79 | 0.81 |
| ET_bf | 0.96 |  | 0.91 | 0.92 | 0.77 | 0.73 | 0.69 |
| joint_bf | 0.97 | 0.93 |  | 0.96 | 0.91 | 0.86 | 0.84 |
| NS-SM_bf | 0.93 | 0.90 | 0.91 |  | 0.84 | 0.77 | 0.78 |
| SMOS_if | 0.63 | 0.63 | 0.59 | 0.64 |  | 0.94 | 0.86 |
| SMAP_if | 0.56 | 0.55 | 0.51 | 0.59 | 0.60 |  | 0.83 |
| S1RT_if | 0.27 | 0.22 | 0.27 | 0.21 | 0.13 | 0.18 |  |

**Figure 6: Correlogram between four baseline and three SM-based inversion frameworks. The analysis is separated between the spatial correlation (red area) of irrigation maps (Figure 7), and the temporal correlation (green area) of monthly irrigation amounts (Figure 5).**

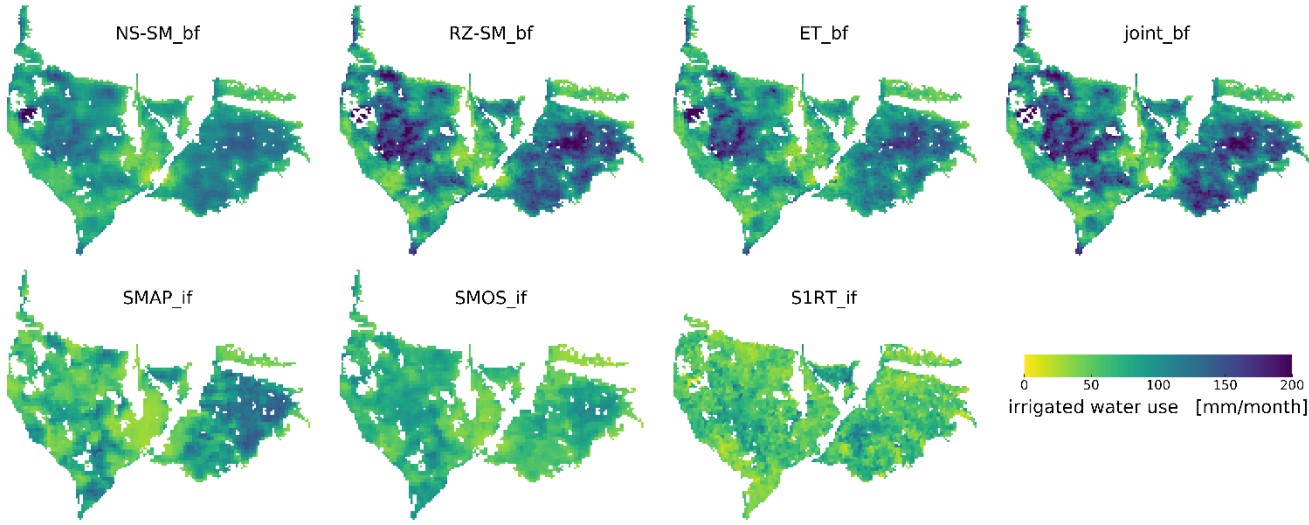


**Figure 7: Mean monthly main irrigation season maps for 2016 and 2017 from four baseline and three SM-based inversion frameworks. The main irrigation season is assumed April – October.**


