# Peer review of "An inter-comparison of approaches and frameworks to quantify irrigation from satellite data"

_Hydrology and Earth System Sciences, 2023_

## Author Response (AR1)

[Reviewer comments in normal font; *Author replies in itialic*]

**REVIEWER 1**

The manuscript provides a welcomed intercomparison of approaches and frameworks to quantify irrigation from remotely sensed soil moisture and ET data. Overall, the results improve understanding of the shortcomings and pitfalls of various approaches designed for this task. I recommend publishing the manuscript after addressing the below comments. This study is likely to inform future irrigation quantification analyses.

***Reply:*** *We thank the reviewer for the positive and constructive feedback that will help us to further improve our work. Below, we outline how we consider responding to the issues pointed out by the reviewer in the revision and what changes we intend to implement.*
***Changes:*** *As we updated some of the figure layouts, we found an error in our calculations for soil moisture residuals where days with less than 75% observation coverage were excluded instead of 50% as described in the paper. We updated our analysis accordingly. Overall, the results are very similar and did not change the interpretation of our results. Only the NS-SM_bf approach performs significantly better on a seasonal basis, which can be seen in Table 3. We changed the reported performance throughout the manuscript and added new text on the NS-SM_bf estimates (lines: 447-449), we updated Table 3, Figure 4, Figure 5, and Figure 7.*

**Primary concern**

Eq. 3: This study calculates irrigation as the sum of SM and ET residuals (satellite – model). This seems like irrigation will often be double counted, because surface soil biases propagate to ET biases. There is a time lag, such that a wetter soil surface (from the satellite) that is attributed to irrigation from a previous time step will be accounted for as irrigation again in a later time step when the water returns to the atmosphere. Is it possible that this could be compensating for other sources of error that favor underestimated irrigation?

If I am missing something about this method that is implemented to avoid double counting, please elaborate on this; if not, please discuss this source of error in the manuscript and how it may be data-specific.

***Reply:*** *Thank you for your comment. The SMOS DISPATCH dataset represents the topsoil (upper 5 cm) soil moisture. This information is difficult to use in the calibration of a hydrological model because the obtained soil moisture could have been obtained during or just after a rainfall event. In this case, most soil water will be in the topsoil, and during the following days, the water will infiltrate deeper into the soil or evapotranspire. On the other hand, if the hydrological model receives the same rainfall input, the model will distribute the added water between all soil layers and outgoing fluxes within a timestep (one day). This makes it challenging to directly compare model estimates and satellite observations of soil moisture. We have tried to overcome this issue by using an equation from Albergel et al 2008 that uses topsoil soil moisture to estimate the amount of water in the root zone*

*which is more comparable with the hydrological model simulations. During this conversion, we have removed a substantial amount (around 40% of the summed soil moisture increase) of the topsoil soil moisture that enters the soil. The amount of water removed is assumed to represent evapotranspiration deeper infiltration and drainage. The conversion to rootzone soil moisture is the main reason why we do not expect the double counting of irrigation water to be a significant part of our results.*

***Plan for revision:** We will make sure that this is better described in the manuscript.*

***Changes:** We have added two new paragraphs to section 3.4 that addresses the main concern about the potential double counting of irrigation water when considering both soil moisture and ET. Overall, due to the substantial removal of near-surface soil moisture input (40%) from the conversion to root zone soil moisture as proposed by Albergel et al. (2008) and the spatial aggregation of daily mean soil moisture residuals at the district scale, we do not expect that our irrigation estimates will be severely affected by double counting of water through the joint analysis of ET and soil moisture residuals. We have tried to better describe the difference between the SMOS DISPATCH near-surface soil moisture and our root zone soil moisture estimate, why the root zone estimate is needed for proper model calibration or rainfed soil moisture, and finally how the use of root zone soil moisture allow us to consider both soil moisture and ET changes despite them being interconnected. (lines: 244-280)*

**Specific comments**

Lines 109, 115: the semicolon should be a colon.

***Plan for revision:** Will be changed in the manuscript.*
***Changes:** The semicolon was changed to a colon (line: 111).*

Line 120: Please add an additional sentence that explains the reasoning/logic for the assumed losses.

***Reply:** Thank you for your comment.*
***Plan for revision:** We will incorporate these reasons into the manuscript.*
***Changes:** We added the rationale behind the assumptions of irrigation losses adopted (lines: 124-128)*

Section 3.1: Can you please provide justification of why MODIS ET is used instead of other products?

***Reply:** We choose the MODIS 16 ET product because it was found to provide valid estimates across European catchments compared to other products (Stisen et al., 2021) and to be influenced by irrigation (Dari et al., 2022; Stisen et al., 2021).*
***Plan for revision:** Make it clear that the choice of MODIS ET was based on other studies' comparisons of different ET products.*
***Changes:** We added that other ET products were also considered before choosing MODIS ET (lines: 144-145).*

Section 3.1: Was a downscaled SMOS product used instead of other high-spatial-res produces (e.g., SMAP-S1) to maintain high temporal resolution? Has there been validation to ensure the SMOS SM product maintains irrigation signals (e.g., similar to Lawston et al., 2017 or Jalilvand et al., 2021)? I believe this is important because the Kumar et al. (2015) analysis seems to suggest the SMOS product fails to detect a significant portion of irrigation signals (which is also related to my concern above regarding Eq. 3).

Lawston, P.M., Santanello, J.A., Kumar, S.V., 2017. Irrigation Signals Detected From SMAP Soil Moisture Retrievals: Irrigation Signals Detected From SMAP. Geophys. Res. Lett. 44, 11,860-11,867. https://doi.org/10.1002/2017GL075733

Jalilvand, E., Abolafia-Rosenzweig, R., Tajrishy, M., Das, N.N., 2021. Evaluation of SMAP-Sentinel1 High-Resolution Soil Moisture Data to Detect Irrigation over Agricultural Domain. IEEE J. Sel. Top. Appl. Earth Observations Remote Sensing 1–1. https://doi.org/10.1109/JSTARS.2021.3119228

Kumar, S.V., Peters-Lidard, C.D., Santanello, J.A., Reichle, R.H., Draper, C.S., Koster, R.D., Nearing, G., Jasinski, M.F., 2015. Evaluating the utility of satellite soil moisture retrievals over irrigated areas and the ability of land data assimilation methods to correct for unmodeled processes. Hydrol. Earth Syst. Sci. 19, 4463–4478. https://doi.org/10.5194/hess-19-4463-2015

*Reply: The SMOS-DISPATCH product was used in this area due to its high spatial and temporal resolution and well-documented ability to detect irrigation in this region (Merlin et al. 2013, Dari et al. 2021). However, the original SMOS SM 40 km product does not reflect the spatial distribution of soil moisture from irrigation, meaning that the spatial soil moisture irrigation signal comes from DISPATCH downscaling that uses MODIS NDVI and land surface temperature. We have analyzed the SMOS-DISPATCH product by comparing climatologies of irrigated and rainfed cropland to ensure that the areas have similar soil moisture content during winter (no irrigation) and diverging content during the irrigation season as expected.*
*Plan for revision: We will add that the irrigation signal in the SMOS-DISPATCH dataset is a product of the downscaling method that redistribute the 40km mean SMOS SM based on MODIS NDVI and land surface temperature.*
*Changes: We added that the distribution of irrigation is introduced to the SMOS dataset through the DISPATCH downscaling algorithm (lines: 159-161).*

In Section 4, abbreviations for the approaches are introduced. It would be useful if in the former section, a table is created that summarizes all evaluated approaches and defines the abbreviations for readers to reference. This would be particularly helpful because the abbreviations are used in Figure legends.

*Reply: Thanks, indeed a very useful table to keep track of the various approaches*
*Plan for revision: Such a table will be added to the manuscript.*
*Changes: A table has been added with an overview of the different frameworks and approaches within this study (line: 323).*

Table 1: Can a figure be added either to main text or supplementary that shows comparisons of the model and observed SM and ET throughout the period? This can be useful to visualize bias characteristic (e.g., random vs. systematic)

**Plan for revision:** *Time series will be added to the supplementary material showing the comparison of observed and modeled soil moisture.*
**Changes:** *We added five figures to the supplementary materials showing rainfed biases from the four model calibrations (Figures 2S-6S). We also have a table showing a couple of bias measures: the mean error (ME) and standard derivation of rainfed residuals. A reference to these results was added to the manuscript (lines: 337-338).*
* * *
Figure 3: please use different colors to differentiate between NS-SM_bf and RS-SM_bf. Please add the benchmark to the legend. Is there a reason the benchmark is shaded instead of a line (like the predictions)?  (Similar sentiments for Figures 4 & 5 aesthetics)

**Reply:** *Thanks for the comment. For figures 3 and 5 we would argue that having different plotting e.g. shaded for benchmark and line plots for results makes it easier to visualize the similarities and differences. In Figure 4 the goal is to compare the calibration approaches which is why we choose the joint approach as a shaded plot behind the separate approaches which makes it easier to show how the contribution from soil moisture and evapotranspiration varies according to the calibration strategy.*
**Plan for revision:** *We will add the benchmark to the legend in Figure 3 and find other colors for NS-SM_bf and RS-SM_bf.*
**Changes:** *The color for the NS-SM_bf graph was changed to black for a better visual effect in Figure 3 and Figure 5. We also changed the legend of Figure 4 to make it easier to interpret.*
* * *
Section 4.2 can benefit from more attention to writing to report results in a more clear and concise manner

**Plan for revision:** *We will try to rewrite parts of the section to make it more concise.*
**Changes:** *Changes have been made in the section to make the paragraphs more similar in the way they are structured regarding describing results and interpretation.*
* * *
Paragraph starting in line 333: Is this bias characteristic model specific to the mHM? If calibration considered other metrics (e.g., NSE) could this error source be reduced?

**Reply:** *We did not test other hydrological models to see whether this behavior is specific to the mHM model, but the mean and standard deviation of simulated soil moisture will always relate to the parameters related to wilting point and saturated water content. Such parameterizations are found across hydrological models. We do not think of this as a model issue, but rather an issue related to the SMOS-DISPATCH datasets mean-variance relationship that is against our expectations, which could relate to uncertainties in the downscaling method. Choosing another calibration metric would not have solved this issue*

*because of the final rescaling between the modeled and satellite mean soil moisture, which was necessary for the model to have a more comparable reaction to the precipitation with respect to the satellite-based soil moisture.*
**Plan for revision:** *None*
* * *
Line 392: comma after "Figure 6" should be removed

**Plan for revision:** *Will be removed from the manuscript.*
**Changes:** *Have been removed. (line: 457)*
* * *
Section 4.3: It would be extremely helpful to the community if this paper summarizes the insights from analyses and comparisons in this section. Namely, a table which summarizes the strengths and weakness (e.g., uncertainty sources) of the approaches, and which approaches are more robust to various uncertainty sources (e.g., precipitation, satellite temporal and spatial resolution, noise, etc.). This table would likely be the primary take-away of the study.

**Reply:** *Thank you for this comment.*
**Plan for revision:** *We will add a table that summarizes some of the outcomes of this study.*
**Changes:** *We have added a new section and table on the influence of different uncertainty sources on the two frameworks. (lines: 497-533)*

[Reviewer comments in normal font; *Author replies in itialic*]

**REVIEWER 2**

This study focuses on comparing different methods for estimating irrigation water use. The topic is important for understanding the human impact on the hydrological water cycle, and there are various methods available with their own advantages and disadvantages. The authors conducted a comparison between a baseline model representing rainfed agriculture and a satellite-based model that captures irrigation. The difference between the two models represents the unmodeled process, which in this case is irrigation. To ensure accuracy, the authors calibrated the model using rainfed pixels to remove biases between the model and satellite observations before calculating the difference. Other irrigation estimation methods, such as the water inversion method using satellite soil moisture and evapotranspiration (ET), are also used and discussed. The paper provides a comprehensive comparison of different methods, but some details are skipped, possibly assuming that readers are familiar with previous related papers. One concern raised is the possibility of double-counting water when estimating irrigation using both soil moisture and ET residuals, as these variables are interconnected. Overall, the paper is recommended for acceptance after addressing the mentioned comment and the below detailed comments.

*Reply:  We thank the reviewer for the positive and constructive feedback that will help us to further improve our work. Below, we outline how we consider responding to the issues pointed out by the reviewer in the revision and what changes we intend to implement.*
*Changes: As we updated some of the figure layouts, we found an error in our calculations for soil moisture residuals where days with less than 75% observation coverage were excluded instead of 50% as described in the paper. We updated our analysis accordingly. Overall, the results are very similar and did not change the interpretation of our results. Only the NS-SM_bf approach performs significantly better on a seasonal basis, which can be seen in Table 3. We changed the reported performance throughout the manuscript and added new text on the NS-SM_bf estimates (lines: 447-449), we updated Table 3, Figure 4, Figure 5, and Figure 7.*

**Major comments**

It appears to me that author assumed the reader has already read their previous papers so they did not explain some terminologies or hypotheses that they had in the abstract, for instance in L15-17 it is not clear what are the satellite or the rainfed framework and what they meant by the baseline framework. Later in the introduction at L43-44, the study hypothesis is explained, I think something to this effect can be added to the abstract.

*Reply: Thank you for your comment.*
*Plan for revision: We will try to make this clearer in the manuscript, especially focusing on the definition of key terminologies upfront in the introduction.*

*Changes: We have added a little more information to the abstract describing the baseline framework (lines: 16-17).*
* * *
I do not understand lines 236-244 regarding how the water is not counted twice. The ET and soil moisture are interconnected, such that an increase in ET results from an increase in soil moisture. Technically some of the water that enters the soil and increased the soil moisture will later be consumed by the plant with a delay and transpired into the atmosphere. Moreover, it is not necessarily from the rootzone, some studies showed that plants' roots get most of the water from topsoil rather than deeper layers. Thus, the water that is once accounted as residual soil moisture will be later extracted by the plant root and then accounted for twice in the calculations.

*Reply: Thank you for your comment. The SMOS DISPATCH dataset represents the topsoil (upper 5 cm) soil moisture. This information is difficult to use in the calibration of a hydrological model because the obtained soil moisture could have been obtained during or just after a rainfall event. In this case, most soil water will be in the topsoil, and during the following days, the water will infiltrate deeper into the soil or evapotranspiration. On the other hand, if the hydrological model receives the same rainfall input, the model will distribute the added water between all soil layers and outgoing fluxes within a timestep (one day). This makes it challenging to directly compare model estimates and satellite observations of soil moisture. We have tried to overcome this issue by using an equation from Albergel et al 2008 that uses topsoil soil moisture to estimate the amount of water in the root zone which is more comparable with the hydrological model simulations. During this conversion, we have removed a substantial amount (around 40% of the summed soil moisture increase) of the topsoil soil moisture that enters the soil. The amount of water removed is assumed to represent evapotranspiration deeper infiltration and drainage. The conversion to rootzone soil moisture is the main reason why we do not expect the double counting of irrigation water to be a significant part of our results.*
*Plan for revision: We will make sure that this is better described in the manuscript.*
*Changes: We have added two new paragraphs to section 3.4 that addresses the main concern about the potential double counting of irrigation water when considering both soil moisture and ET. Overall, due to the substantial removal of near-surface soil moisture input (40%) from the conversion to root zone soil moisture as proposed by Albergel et al. (2008) and the spatial aggregation of daily mean soil moisture residuals at the district scale, we do not expect that our irrigation estimates will be severely affected by double counting of water through the joint analysis of ET and soil moisture residuals. We have tried to better describe the difference between the SMOS DISPATCH near-surface soil moisture and our root zone soil moisture estimate, why the root zone estimate is needed for proper model calibration or rainfed soil moisture, and finally how the use of root zone soil moisture allow us to consider both soil moisture and ET changes despite them being interconnected. (lines: 244-280)*
* * *
L235-236 & L304-305: If the soil is over-irrigated then the surface soil will become saturated and SM will stay at a constant level. Consequently, it would not be able to reflect both soil moisture storage and ET fluxes change. Please comment on this.

**Reply:** *It's true that over-irrigated fields would pose an issue and require that drainage and overland flow would also need to be accounted for. However, based on the following plots showing SMOS DISPATCH soil moisture observations for a random set of irrigated pixels, we draw the conclusion that this is not a big concern within our study area. Results from Dari et al. 2020, who used a soil moisture-based inversion framework, also suggested that drainage from irrigation accounted for less than 0.5% of the irrigation within our study area.*

[Figure]

*The six plots show the SMOS DISPATCH soil moisture observations (blue) together with the modeled baseline for the RZ_SM_bf approach over two years from 2016-2017.*

**Plan for revision:** *We will mention that a drainage and overland flow term might be necessary to consider if this method were to be used in an area with known over-irrigation.*
**Changes:** *We have added a sentence describing how to deal with over-irrigation in terms of other variables (lines: 279-280)*

**Minor comments**

Figure 2) There are some positive and negative residuals after calibration for the rainfed cropland that can propagate to the residual estimated over the irrigated pixels, how are these errors treated in your approach?

**Reply:** *The baseline model does sometimes either over- or underestimates the rainfed soil moisture or evapotranspiration which will affect the irrigation estimates. However, the overall mean error of the baselines is close to zero for all calibration targets within the four approaches, which implies that the error related to the baseline uncertainty will even out over the simulation period.*
**Plan for revision:** *We will add rainfed time series to the supplementary material from the calibration to illustrate the small bias. The overall SM and ET biases over rainfed land will be quantified as well as the standard deviation of the rainfed residuals.*
**Changes:** *We added five figures to the supplementary materials showing rainfed biases from the four model calibrations (Figures 2S-6S). We also have a table showing a couple*

*of bias measures: the mean error (ME) and standard derivation of rainfed residuals. A reference to these results was added to the manuscript (lines: 337-338).*
* * *
L193: Could you add a figure with 4 maps to the manuscript? Firstly add two maps derived from ET and soil moisture temporal stability analysis. Secondly, create an overlap map that combines ET and SM maps. Then compare it with an independent land use map that shows the rainfed and irrigated cropland and report how accurate was the rainfed cropland mapping. This step is crucial as the bias removal process is conducted based on the selected pixels from these maps.

**Reply:** *Thank you for this comment.*
**Plan for revision:** *We will add two maps to the supplementary materials showing the results from the ET and soil moisture temporal stability analysis separately. The combined results can be seen in Figure 1 compared with an aerial photo from where there is a clear difference between the green irrigated and beige rainfed cropland.*
**Changes:** *We added two maps to the supplementary materials with the results from the temporal stability analysis from ET and soil moisture, respectively (Figure 1S). Also, a small section was added to describe the results. A reference to these results was added to the manuscript (lines: 207-208)*
* * *
L197: By calculating the MAE spatially at each time step, you won't have individual values for each pixel. Instead, there would be only 10 or 14 values for the calibrating parameters after optimization. However, wouldn't it have been more beneficial to have separate values for each pixel by calibrating for all the pixels during the non-irrigated period?

**Reply:** *Thanks, we did optimize by calculating MAE for all pixels during the two-year calibration period. Indeed it would have been optimal to calibrate the model over both rainfed and irrigated cropland during the non-irrigated period, but we know that some irrigation does occur within the districts throughout the year which is why we decided only to include exclusively rainfed cropland in the calibration.*
**Plan for revision:** *We will try to better describe how we calculated the MAE and choice of a target area.*
**Changes:** *We have added comments on how MAE was calculated and the choice of target area for the calibration (lines 208-209).*
* * *
In equation 1, how time is considered in the calculation of BAE are you again averaging MAE for all time steps? If yes, please show that in the equation and also mention this in the text.

**Reply:** *The mean absolute error (MAE) is calculated on a pixel level and not by using an area average like the irrigation estimates.*
**Plan for revision:** *We will make this more clear in the manuscript.*

*Changes: We have added comments to how MAE was calculated (line: 210)*
* * *
L205-210: I am having trouble understanding the steps described in L205-210. Are you implementing both model calibration through optimizing an objective function and rescaling by adjusting the model mean SM to match the satellite SM mean? Is rescaling necessary or has it already been accounted for in the calibration process?

**Reply:** *Yes, we are implementing both in the calibration process due to a model/observation issue discussed in lines 333-334.*
**Plan for revision:** *We will make sure to make this more clear in the manuscript.*
*Changes: We have added comments on the calibration strategy to specify that rescaling was used in each iteration (line: 219). Also included how the rescaling affects the calibration (lines: 222-224).*
* * *
L236: replace "it" with water

**Plan for revision:** *Will be changed in the manuscript.*
*Changes: The entire paragraph was changed.*
* * *
L270: what is the rootzone soil moisture data you used for calibration?, mention it here for RZ_SM_bf

**Reply:** *The root zone soil moisture is calculated from the SMOS DISPATCH topsoil soil moisture product by using an equation from Albergel et al. 2008 that estimates the amount of entering water into the soil that represents the root zone after two days from entering the soil.*
**Plan for revision:** *We plan to add a table that summarizes all the models and different inputs for a better overview.*
*Changes: A table has been added with an overview of the different frameworks and approaches within this study (line: 323).*
* * *
Figure 4) I find it a little bit confusing to interpret the legend in Figure 4 legend. To enhance the legend clarity I suggest putting the ET and SM of each approach in individual boxes and labeling them with the corresponding approach names. This adjustment would make it easier to comprehend the legend. Additionally, the solid blue line is not explained in the figure caption.

**Reply:** *Thanks for your suggestion.*
**Plan for revision:** *We will try to make the legend more clear.*
**Changes:** *We changed the Figure 4 legend to make it easier to interpret (line: 780).*
* * *
L315-316: How can we ensure that storage of the water from the previous season is the primary reason for estimating a higher irrigation value compared to the benchmark and not an overestimation of irrigation by the model?

**Reply:** *We don't have any data to assure that some of the estimated irrigation is water stored from last season. However, in Algerri Balaguer we know that this is common practice and we know that reservoirs exist within all irrigation districts which allow the farmers to distribute the water as needed. If we look at Figure 3, the use of storage water in Algerri Balaguer in April 2017 is clear, but not within any other districts. If this irrigation peak were caused by a large model bias, we would expect a similarly large increase within the surrounding districts.*
**Plan for revision:** *We will comment further on this in the manuscript.*
**Changes:** *Nothing were changed, we choose only to include this answer in the reviewer reply.*
* * *
L325: Perhaps the reservoir is operating based on a relatively fixed plan, regardless of how much precipitation is received. Have you explored this possibility?

**Reply:** *Yes, we also think that makes sense that large reservoirs like this work with a relatively fixed schedule based on the reservoir capacity, incoming water, and when irrigation water normally is needed within the districts. We do not have contact with the reservoir managers to get this confirmed.*
**Plan for revision:** *Add a comment on the possibility of a fixed irrigation schedule to explain benchmark data in relation to the actual irrigation practice.*
**Changes:** *We added a line commenting on different possibilities why the benchmark data are looking rather similar despite meteorological variability (line: 383).*